# Brief Communication: Solar Radiation Management not as effective as $CO_2$ mitigation for Arctic sea-ice loss in hitting the 1.5°C and 2°C COP climate targets.

Jeff K. Ridley, Edward W. Blockley

Met Office, Exeter, EX1 3PB, UK

*Correspondence to*: Jeff Ridley (jeff.ridley@metoffice.gov.uk)

**Abstract.** An assessment of the risks of a seasonally ice-free arctic at 1.5 and 2.0°C global warming above pre-industrial is undertaken using model simulations with solar radiation management to achieve the desired temperatures. An ensemble, of the CMIP5 model HadGEM2-ES uses solar radiation management (SRM) to achieve the desired global mean temperatures. It is found that the risk for a seasonally ice-free Arctic is reduced for a target temperature for global warming of 1.5°C (0.1%) compared to 2.0°C (42%), in general agreement with other methodologies. The SRM produced more ice loss, for a specified global temperature, than for $CO_2$ mitigation scenarios, as SRM produces a higher polar amplification. .

## 1 Introduction

The 21st Conference of Parties (COP) to the UN Framework Convention on Climate Change held in Paris in 2016 made a commitment to limiting global-mean warming since the pre-industrial era to well below 2.0°C and to pursue efforts to limit the warming to 1.5°C (UNFCCC, 2015). The 1.5 °C target reflects a threshold at which the likely local impacts of climate change are beyond the ability of society to cope with. This is especially applicable to the small island states that are susceptible to sea-level rise, ground-water salinification and loss of coral reefs. One such risk is the loss of Arctic sea ice, for which previous studies (Sanderson et al., 2017; Screen & Williamson, 2017; Jahn, 2018; Niederdrenk & Notz, 2018; Sigmond et al., 2018) used a number of methodologies with various climate models under $CO_2$ mitigation scenarios. The findings are broadly similar, that there is a low chance of an ice–free Arctic if global temperatures are limited to 1.5°C and a moderate chance at 2°C.

It has been suggested that geoengineering, otherwise known as solar radiation management (SRM), may be a stopgap measure to halt these impacts, stabilising Earth's temperature at 1.5 K, before $CO_2$ mitigation can take effect (Chen & Xin, 2017). Here we evaluate the impact of SRM on Arctic sea ice decline and compare with mitigation methods alone, through the implementation of SRM, in our climate model HadGEM2-ES. We use the SRM strategy of stratospheric aerosol injection, which mimics large volcanic eruptions (Crutzen, 2006).

Sea ice hits its smallest extent sometime in September and since the satellite record began in 1979, the Arctic sea ice cover in the month has declined by around 11% per decade (Comiso et al., 2017).. Such a sharp drop off in sea ice has prompted the question of when the Arctic will first see an ice-free summer. By "ice-free" we mean a sea ice extent of less than one million square kilometres, rather than zero sea ice cover. We make this choice because although the central Arctic Ocean is free of ice, the thick ice along the North coast of Greenland can take some further decades to melt.

The impacts of a seasonally ice-free Arctic include increased ice loss from Greenland (Day et al., 2013; Lui et al., 2016), and hence sea level rise, and may contribute to extreme weather events in the northern mid-latitudes (Overland et al., 2016; Francis et al., 2017). Furthermore, storms and waves in the open water may cause coastal erosion, impacting marine ecosystems, infrastructure and local communities (Steiner et al., 2015; Radosavljevic et al., 2016).

With the objective to limit the increase in global average temperature to well below 2.0°C above pre-industrial levels and to pursue efforts to limit the temperature increase to 1.5°C above pre-industrial levels, we need to ascertain the costs of mitigation and associated climate risks. It has been suggested that SRM may be a means to reduce the immediate costs of climate mitigation, especially to reach the 1.5C target (Sugiyama et al., 2017). There have been a number of proposed mechanisms to

reduce the solar radiation reaching the Earth's surface through geoengineering (Shepherd, 2009; Ming et al., 2014). Here we employ the SRM methodology of increasing sulphate aerosols in the stratosphere, which in the CMIP5 climate model HadGEM2-ES, is achieved through uniformly increasing the number density of volcanic aerosols. This work expands on the methodology of Jones et al. (2018) where SRM is applied in HadGEM2-ES. Although this method can stabilise global

temperatures, it produces a spatial temperature pattern with over-cooling in the tropics and slight warming at high latitudes (Kravitz et al., 2017). This means it may be effective in reducing ice loss compared to doing nothing, as SRM cools everywhere, but not so effective compared with reducing greenhouse gas emissions. Here we evaluate this by comparing a geo-engineered 1.5 and 2.0°C worlds with the equivalent temperature $CO_2$ mitigated worlds. The use of modelled SRM in this paper does not endorse or advocate either testing or actual implementation of geoengineering. Our purpose here is to study and inform.

**2 Method**

HadGEM2-ES is a coupled AOGCM with atmospheric resolution of N96 (1.875°×1.25°) with 38 vertical levels and an ocean resolution of 1° at mid-latitudes (increasing to 1/3° at the equator) and 40 vertical levels (Jones et al., 2011). The ocean grid has an island at the North Pole to avoid the singularity caused by a convergence of the meridians. The sea ice component uses elastic-viscous-plastic dynamics, five ice thickness categories, and zero-layer thermodynamics (McLaren et al., 2006). The

HadGEM2-ES simulation produces a good representation of Arctic sea ice, thickness, trends, seasonal cycle and variability, when compared against observations (Martin et al., 2011; Baek et al., 2013; Huang et al., 2017). Simulated temperature changes are referenced against the mean global temperature from a 400 year section of a pre-industrial control simulation with constant forcing at 1860 levels of greenhouse gases.

The objective is to explore several SRM scenarios branching from the transient simulations of Representative Concentration

Pathway (RCP) scenarios (van Vuuren et al., 2011). RCP scenarios start from the year 2005 and continue to 2100. The mean of the four RCP2.6 scenario simulations reaches a peak global mean temperature of +2°C while that of RCP4.5 reaches +2.9°C. Each scenario is allowed to develop without SRM adjustment until a global temperature of +1.5°C is reached in RCP2.6 (year 2020), +2.0°C and +2.5°C in RCP4.5 (years 2040 and 2060 respectively in the ensemble means). New simulations, using SRM, are started from these points using continuous injection of $SO_2$ into the model stratosphere between 16 and 25 km. This

$SO_2$ is oxidised to form sulphate aerosols that reflect incoming solar radiation and thus cool the climate. As HadGEM2-ES does not have a well-resolved stratosphere, $SO_2$ is injected uniformly across the globe to reduce any problems with stratospheric transport. Since the global temperature of the RCPs varies in time, the $SO_2$ required to maintain a constant global temperature will also vary in time. The difference between the RCP ensemble mean and target temperatures (e.g. 2.0°C), calculated at 10-year intervals, was used to determine the time-profile of $SO_2$ injection in combination with calibration

simulations to assess the amount of cooling for a given level of $SO_2$ injection (-0.115 °C/Tg [$SO_2$] yr$^{-1}$). Provided the temperature differences are small, a single value for climate sensitivity to $SO_2$ may be applied. The same $SO_2$ time profile was injected into each of the ensemble members with the same target temperature.

For CMIP5 a historical + scenario initial condition ensemble of four HadGEM2-ES members was completed. A larger ensemble is required to generate a probability distribution of sea ice decline. To achieve this we take the four separate RCP

ocean and atmosphere start conditions and intermix them (e.g. RCP ensemble member-1 atmosphere with RCP ensemble member-2 ocean) to provide 16 perturbed members for each start date of 2020, 2040, and 2060. The application of a random atmosphere on an ocean state equilibrates within a few days (Griffies & Bryan, 1997). The resulting ensemble spread in global mean temperature is larger than that for the initial 4-member ensemble, indicating that the resulting initial perturbations are sufficient to generate a wide range of climate trajectories.

The ensembles analysed in this study are as follows:
- Ensemble-1 : starts at 1.5°C on RCP2.6 and levels out at 1.5°C above pre-industrial control (1860).

- Ensemble-2 : starts at 2.0°C on RCP4.5 and levels out to 1.3°C above pre-industrial control (1860).
- Ensemble-3 : starts at 2.5°C on RCP4.5 and levels out to 1.7°C above pre-industrial control (1860).

## 3 Results

The September sea ice extent in the three ensembles (Figure 1) remains stable in Ensemble-1 but recovers in Ensemble-2 and Ensemble-3. The recovery is in line with the downward drift in global mean temperatures as indicated by the reversibility and decadal temperature sensitivity of Arctic sea ice change (Ridley et al., 2012). The spatial pattern of sea ice edge (Figure 2) shows that the model represents a low ice extent for present day in the Greenland Sea when compared with observations. This is because the ice modelled in HadGEM2-ES, in common with many CMIP5 models (Stroeve et al., 2014), is thin in the Atlantic sector and too thick in the Beaufort Gyre, consequently the sea ice retreats in the Atlantic sector with global warming. The ice edge, at equilibrium, is nearly identical in Ensemble-1 and Ensemble-2, with ice retreating further in the Atlantic sector. Meanwhile Ensemble-3 has members with discontinuous ice cover, with a patch of ice in the Beaufort Gyre, where the ice was originally too thick, and extending along the North Greenland and Canadian Archipelago coasts. That Ensemble-3 has a different spatial pattern of the ice edge, and yet is only a few tenths of a degree warmer than the other two ensembles at 2100, associated with the 15% threshold used to derive the ice edge. The summer ice cover in the central Arctic has an extensive marginal ice zone and so the threshold definition of the ice edge at 15% ice concentration is noisy.

The time-drift in September ice extent in Ensemble-2 and Ensemble-3 leads us to conclude that attempting to create a mean state for specific global temperatures, without precise tuning of the SRM for each RCP, is not sensible. Instead, all ensembles are combined to form a continuum of annual global temperature and September Arctic sea ice states. The scatter-plot of all 48 ensemble members and 2880 simulated years is shown in Figure 3. It is expected that the use of SRM will change the regional energy budget, with many models showing an enhanced warming in the Arctic (Kravitz et al., 2017; Jones et al., 2018). To compare SRM and greenhouse gas (GHG)-scenarios for the same global temperature rise, in addition to the SRM ensembles, the data from the transient RCP2.6 and RCP4.5 is added to the scatter-plot. The RCPs climate is moderated by greenhouse gas emissions, and so serve as a reference for the SRM ensembles. The data from the transient simulations shows broadly similar characteristics to the ensemble members, with high scatter in sea ice extent at low global temperature and less at higher temperatures. However, it is evident that the RCP simulations show a marginally greater sea ice extent than for SRM, and we assess this through model polar amplification. The polar amplification, as defined by $\Delta T_{(60\text{-}90°N)}//\Delta T_{(global)}$ (where $\Delta T$ is a 20 year time mean temperature rise - in this case a global rise of 1°C), is 2.48±0.08 for the RCPs and 2.89±0.12 for the SRM ensembles. The higher polar amplification for the SRM case is in agreement with Kravitz et al.(2017). In principle, the higher SRM polar amplification should result in a faster decline of the Arctic sea ice, so we investigate if the sea ice extent is lower for SRM then RCPs at 1.5°C. The mean sea ice extent in the temperature band 1.5±0.1°C (Figure 3a) above preindustrial, is 2.45±0.02 x $10^6$ km$^2$ with SRM and 2.90±0.09 x $10^6$ km$^2$ in the RCPs (with $CO_2$ mitigation). This result shows a higher sea ice loss in the SRM experiments than with mitigation at 99.7% confidence.

The probability distribution function (PDF) is derived for sea ice extent within temperature bands; 1.5±0.1 (sample size 1068 of which 77 are RCP) and 2.0±0.1°C (sample size 341 of which 112 are RCP) above preindustrial. The probability of a single year with an ice extent less than one million square kilometres at +1.5°C is 0.1% and that at +2.0 °C is 42%.

## 4 Conclusions

Similar to previous studies we find a significantly reduced risk of a seasonally ice-free Arctic with a target temperature for global warming of 1.5°C (0.1%) than for 2.0°C (42%). The approach described here differs from other studies which use climate mitigation to limit global temperature (Sanderson et al., 2017; Screen & Williamson, 2017; Jahn, 2018; Niederdrenk & Notz, 2018; Sigmond et al., 2018), and who report broadly similar probabilities. Here, $CO_2$ is allowed to increase and the

global mean temperatures are limited by SRM. We show that, as a result, the Arctic sea ice declines faster using SRM than for an equivalent global mean temperature under GHG mitigation scenarios (RCP) The internal variability of Arctic sea ice is high at 1.5°C, but because of the size of our ensembles we can show a significant difference between SRM and RCP. In common with the studies of Haywood et al. (2013), Jones et al. (2017), Jones et al (2013) and Trisos et al (2018) our study

provides another cautionary aspect for SRM implementation. These studies showed counterbalancing deleterious impacts on Sahelian drought and N. Atlantic hurricane frequency if SRM were applied in a hemispherically asymmetric manner, and a significant termination effect that ecosystems may not have the capacity to deal with should high levels of SRM be relied on. An increased localised SRM over the Arctic can reduce the albedo feedback, but enhances other positive feedbacks from clouds and poleward heat transport. However, sufficient local SRM can halt sea ice decline (Tilmes et al., 2014).Here we show

that SRM is not as effective as conventional mitigation in reducing Arctic sea-ice loss, due to a higher polar amplification for SRM for the same amount of global warming.

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

Figure Captions

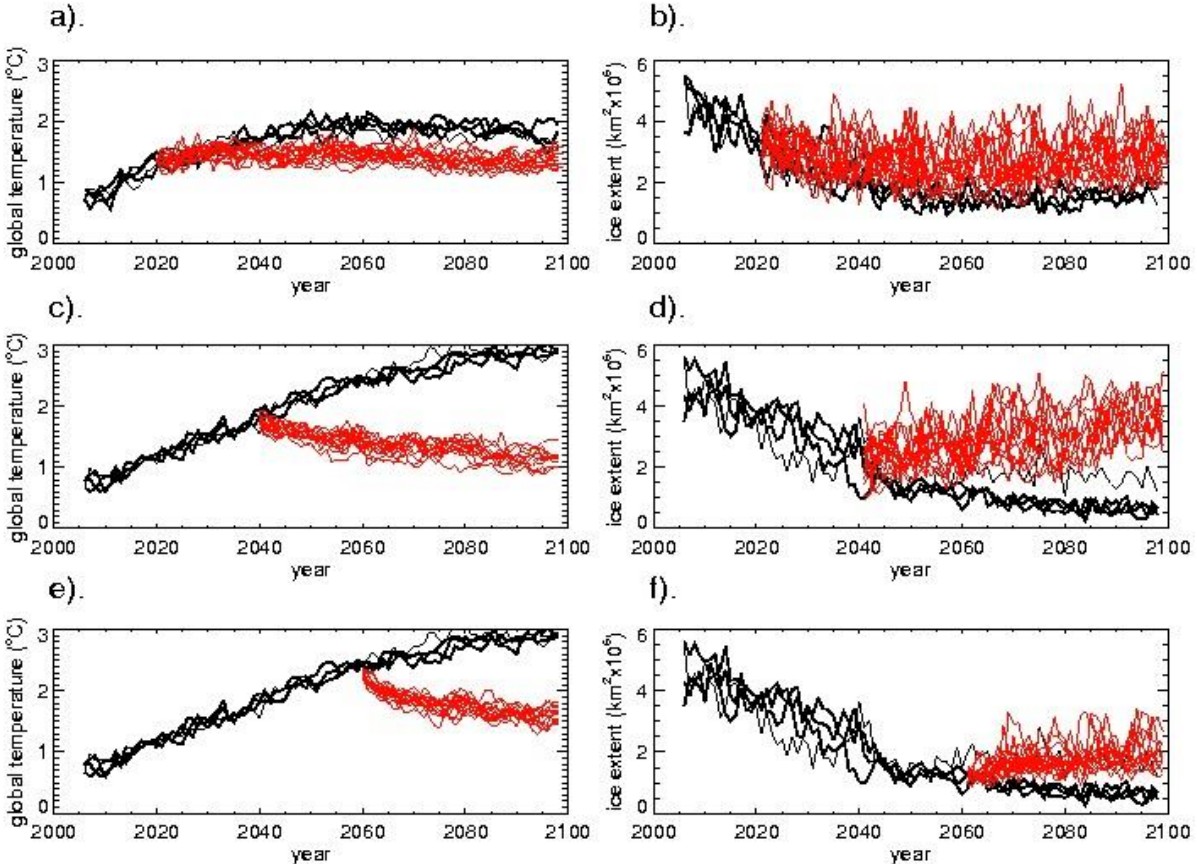

Figure 1 (a) Global mean 1.5m temperature and (b) Arctic sea ice extent for the four ensemble RCP2.6 simulations (black) and the 16 member ensemble-1 initiated from +1.5°C (red). (c) Global mean 1.5m temperature and (d) Arctic sea ice extent

for the four ensemble RCP4.5 simulations (black) and the 16 member ensemble-2 initiated from +2°C (red). (e). Global mean 1.5m temperature and (f) Arctic sea ice extent for the four ensemble RCP4.5 simulations (black) and the 16 member ensemble-2 initiated from +2.5°C (red).

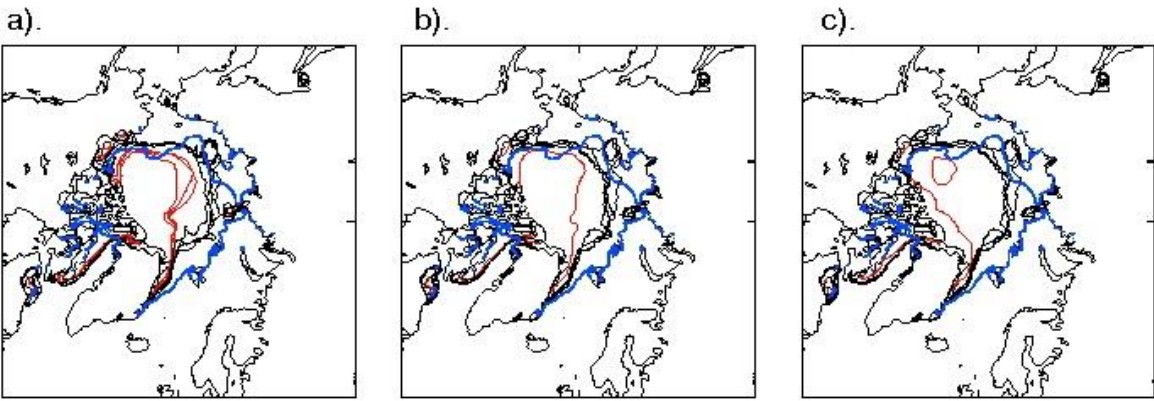

Figure 2. The spatial pattern of the sea ice edge (the 15% concentration contour) with the observations mean for the period 2006-2015 from HadISST (Rayner et al., 2003) in blue, the four-member model RCP ensemble for the equivalent period in black and the 16 member ensemble simulations for the mean of years 2080-2099 in red. (a) The RCP2.6 simulations and ensemble-1; (b) the RCP4.5 simulations and ensemble-2; (c) the RPC4.5 simulations and ensemble-3.

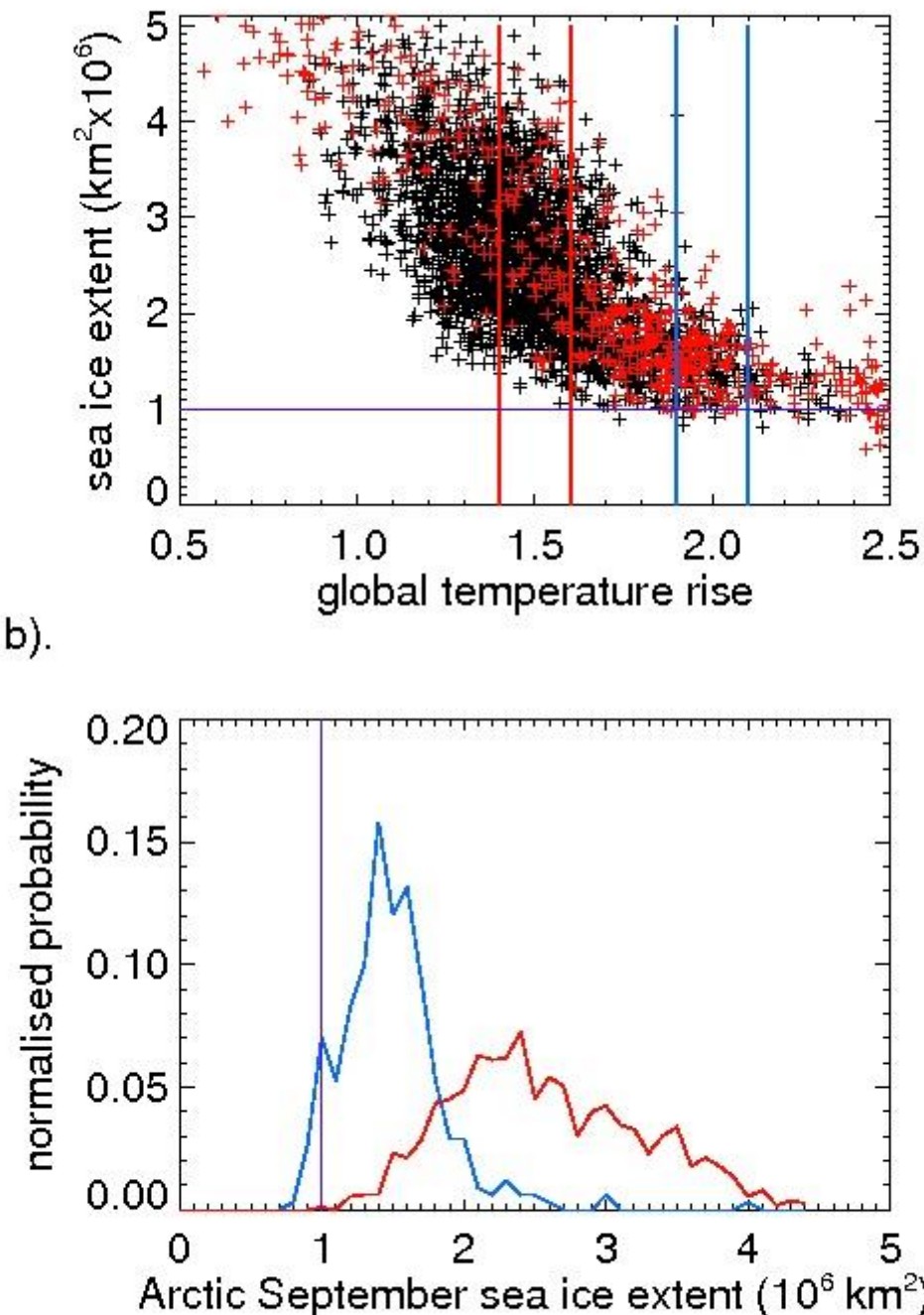

Figure 3. (a) All 48 ensemble members are combined to derive a September ice extent vs global temperature scatterplot (black symbols) with the complete four member RCP2.6 and four member RCP4.5 simulations included (red symbols). The threshold of one million square kilometres signifying an almost ice-free Arctic is shown with the purple horizontal line. The data points used to evaluate the probability distribution function of (b) are selected from the global temperature thresholds of 1.5±0.1°C (red vertical lines) and 2.0±0.1°C (blue vertical lines). (b) The normalised probability distribution functions of Arctic sea ice extent at global temperature rises of 1.5±0.1°C (red) and 2.0±0.1°C (blue) associated with the ensemble members shown in (a). The one million square kilometre threshold for an ice-free Arctic is indicated by the purple vertical line.