# Peer review of "Brief Communication: Solar Radiation Management not as effective as CO2 mitigation for Arctic sea-ice loss in hitting the 1.5°C and 2°C COP climate targets."

_The Cryosphere, 2017_

## Referee Comment (RC1) · Anonymous Referee #1 · 10 Jan 2018

The paper is short, well written and scientifically sound.

The main concern I have is to clarify what the actual contribution of the paper is.

The paper should be better positioned wrt state of the art. In particular the intro should make clear what were the findings from Screen and Williamson (2017) and Sandersen et al. (2017), how the present study differs from those, and how the present methodology brings something different from / completes these studies.

If a clear added value can be defended, then the paper can be published nearly as is.

I would add that, if the contribution is an independent reevaluation of the likelihood of an

ice-free Arctic under 1.5, 2 and 2.5° C targets, using an alternative method (ensemble vs multi-model vs emulator), I'm quite supportive for the paper to be published, even if the final result duplicates previous findings. Independent, repeated tests are in my view as important as original studies.

In practice, this would probably mean moving material from the end of the paper to the end of the introduction, and complete what is only being suggested at the moment by being more explicit.

More specific comments below.

- The advantages / specificities of the SRM method should be clearer and the reason why it has been chosen as well.

- I find the methodology not fully clear. In particular the story of the time dependence of SO2 emissions. Could you illustrate or better describe how SO2 emission depends on time ? Is this constant then stabilised ? Is it ramped up ? Is it non-linear ?

- It is well known that the rate of Arctic sea ice decrease depends on mean state, in particular ice volume. Do you expect a model with less volume and the same experimental setup to give higher probability of sea ice volume loss at 1.5 °C ?

- p. 1 l. 23 I would say that there is a net increase in winter growth because ice is thinning (Bitz and Roe, 2004), but I'm not sure which effects dominates. You should come up with more references or more arguments (for instance a mass balance study in CMIP-X).

- p.1 l. 22. "With global, and regional, warming" sounds weird to my ears.

- p. 1 l. 28 "increased" instead of "increase"
- p. 1 l. 29. I think the increase in extreme weather due to reduced sea ice is quite challenged, in particular the quite convincing study of Blackport and Kushner J. Clim 2016.

- p. 2 l. 7. Replace "this is because" by "we make this choice" or reference others to clarify whether you propose this or whether this is standard practice.

- p. 2 l. 20. Explain why you use this method.

- One inconsistency is how °C is spelt. Sometimes without the °, sometimes with space, sometimes not. Make it consistent.

---

## Referee Comment (RC2) · Anonymous Referee #2 · 31 Jan 2018

This brief communication shows results on the likelihood of an ice-free Arctic under the low-warming IPCC targets, using experiments with solar radiation management with the HadGEM2-ES model. It shows that limiting the global temperature rise to around 1.5°C reduces the likelihood of an ice-free Arctic to 0.2%, compared to 43% for warming around 2°C. This study is interesting and shows similar results to previous work using other models, in particular with the CESM in Sandersen et al. (2017) and the CMIP5 RCPs in Screen and Williamson (2017). This shows that those results are robust across different models. So while the results are not new, they are worthwhile to be published, in particular in light of the upcoming IPCC report on the low warming targets. However, I find the submitted manuscript not publishable in its current form. Even

for the category "Brief communication" I find the results section too short – it is shorter than the methods and about the same length as the conclusions and introduction individually. This paper really seems to be pushing the "least publishable unit" into the unacceptable category, and I can not support that. There is lots of additional analysis that could easily be done with these simulations, even expanding directly on the figures shown. For example Figure 2 is not discussed much and that could be expanded to add new insights not previously shown to this discussion (the spatial aspect of the ice edge under these targets). That said, I have a long list (as long as the paper. . .) of things that need to be addressed, detailed below. Furthermore, the article has many typos, fuzzy figures, and imprecise and confusing statements and captions. The figures also need to be revised. Overall, I see merit in publishing these results as brief communication, but not in the present form, and therefore recommend major revisions.

Detailed comments:

Page 1, Line 8-9: Why would we want to "reduce the internal variability" if we want to produce a probability density function of an ice-free state? Is this a typo, and it should be "deduce"?

Page 1, line 18: Seems to be missing a noun at the end of the sentence? Should be "one such system"

Page 1, Line 25: Should be "uptake", not "up-take"

Page 1, Line 28: Should be "increased", not "increase"

Page 1, Line 3: Why "then" here, doesn't really make sense, as it does not relate to the previous sentence ("Sea ice then hits its smallest extent sometime in September . . .")

Page 2, Line 13: two .. at the end of the section

Page 2, Line 13: I think the introduction needs to reference the oprevious work on this topic, and not make it sound like this hasn't been done yet (Screen and Willamson 2016, Sanderson et al. 2017)

[Figure]

Page 2, Line 15: Degree symbol is incorrect, should be ° not âŮę . This is correct for °C, but not for degree related to N/S.

Page 2, Line 30 to Line 2 Page 3: Is there just one simulation for each RCP? Sounds like it, since just one year is given for reaching these target temperatures. But then it says "For CMIP5 a historical + scenario initial condition ensemble of 4 HadGEM2-ES members was completed.". So does the ensemble mean of these four cross the threshold at that time? That needs to be made clearer. The figure 1 only shows one line for each RCP. But then says that that is the ensemble mean of 4 RCP members. But then each member of the 16 member ensemble for reduced temperatures is shown. Why is that done? It makes no sense to me. The 4 ensemble members for each RCP should be shown as well, so it is clear whether the new ensemble is outside the internal variability of the initial RCP. That would help to substantiate the statement on page 3, line 5 that (The resulting ensemble spread in global mean temperature is larger than that for the initial 4-member ensemble). Not that I doubt that, but it is always nice to see this, and to see how much larger the spread is. Adding the ensemble members for the RCPs would also explain why some of the red lines start well above or below the black line.

Page 2, Line 30 to Line 2 Page 3: What is the reference period for the 1.5C and 2C and 2.5C warming? That's not mentioned and various papers use different periods.

Page 3, line 3-6: This seems like quite a large perturbation, mixing the ocean initial conditions with an atmosphere that it isn't equilibrated with it. Has this approach been tested before? I have only heard of others using much smaller initial condition perturbations.

Page 3, line 7-10: Why were these target temperatures of 1.3 and 1.7C chosen, when the IPCC target is 1.5C?

Page 3, Results section: This is a VERY short result section, shorter than the methods and about the same length as the conclusions. That seems to me to be very

much below the "least publishable unit" standard, and while interesting, I can't support publishing research in such minimal increments. There is lots of additional analysis that could easily be done with these simulations related to Arctic sea ice to make this acceptable, or even expanding the current description of the results. I know this is a "Short communication", but even for that I find the results too brief.

Page 3, line 29: Extra "In" at start of sentence

Page 3, line 30: Why is the "quantitative result described here" only listed a paragraph later? That makes it hard to compare it with the other studies. Which should be discussed in the introduction already, in my opinion.

Page 4, line 4: Should be "to", not "do"

Page 4, Line 1-7: Didn't Screen and Williamson (2017) look at the probability of just one ice-free Arctic, while here and in Sandersen et al. (2017) the overall probability is assessed?

Page 4, Conclusions in general I think these need to be more carefully written. I don't think the statement "since the Arctic sea ice in CMIP5 models is effectively in equilibrium with the instantaneous global temperature" is defensible, as that would mean that Arctic sea ice in a given year is dependent on the global temperature. The cited work shows that this is true in the long-term sense, but not for year-to-year variability. Furthermore, this doesn't apply to all properties of Arctic sea ice, so this needs to be more precisely formulated (September, extent).

Generally: Please use a consistent number of significant digits, and not 1.5C and 2.5C but then 2C.

References: Please check these for inconstant formatting – not my job as a referee to fix those

Figure 1: As mentioned earlier, this figure should either show the ensemble mean or the individual members, not a mix of both. Unless there is a good reason for that, but

none is articulated. Furthermore, the figure is fuzzy, so it doesn't seem to be saved in a vector format. Need to be switched out to be acceptable.

Figure 2: It is not clear to me what this figure really shows. First of all, this figure does not show the "spatial pattern of the Arctic sea ice extent", but the "spatial pattern of the sea ice edge". Which is only the 15% contour, and hence is very noisy. Furthermore, it is unclear to me what both the red and black lines are. It says early on that it shows the "mean of years 2080-2099", for certain temperature thresholds in certain simulations. But then it goes on to say it shows "the mean (years 2006-2025) of the four" RCPs. I also don't understand "starting at +1.5°C above preindustrial in RCP2.6 (left), +2°C in RCP4.5 (center) and +2.5°C in RCP4.5 (right)." This clearly needs to be rewritten to make sense to a reader who hasn't made the figure. It says the red lines are for the 16 individual ensemble members, but those are not the RCPs. Going back to how the ensembles are defined, I think that's what the red lines show, but then these are not for temperatures 1.5C, 2.0C and 2.5C, but for 1.5C, 1.3C and 1.7C? Panels also should have labels (a) and (b) and (c) for ease of reading/referencing. Furthermore, based on what I see, the black line in each panel is different. I assume that is because it is for the different temperature thresholds in the two RCPs mentioned above, and probably for 2006-2025, as they look like present-day. But why use different baselines for each, and what is the third one, since there are only two RCPs? That all makes no sense to me. It would make more sense to use the present-day period for all of them, potentially both for the model and observations, to show how realistic the model is. And then show the ice edge for the 2080-2099 for each of the three ensembles. But since their temperatures aren't that different, and are lumped together in the next figure, why they should be shown is unclear. To show the difference between the ones that end at 1.5 and 1.3 degree C versus ensemble 3 that ends at 1.7 degree C?

Figure 3: This figure is also fuzzy and out of focus. It needs to include a higher resolution figure to be considered for publication. Furthermore, it is unclear what exactly it shows. The caption needs to state what the size of the boxes is (+/- 0.1 degree C

around the mentioned thresholds, according to my reading of the graph), as it does not show the probability right at the quoted temperature thresholds. Otherwise, this figure is the most interesting one. I do wonder how it compares to the transient RCPs with that model though, and hence how much it adds to just using those. Can those be added here? They all cross the same temperature range, so could be included in the PDF. Currently it seems to only include the 3 ensembles with 16 members each (3*16=48). That leads to many fewer members that are in the 2 degree warming box than in the 1.5 degree warming box, and hence could be influencing the probability distributions shown in panel 2. Adding the RCPs in here would help to rectify this. Panels also should have labels (a) and (b)

---

## Author Comment (AC1) · 16 Mar 2018

**Response to Anonymous Referee #1**

We thank the reviewer for their useful comments

The paper is short, well written and scientifically sound.

The main concern I have is to clarify what the actual contribution of the paper is. The paper should be better positioned wrt state of the art. In particular the intro should make clear what were the findings from Screen and Williamson (2017) and Sandersen et al. (2017), how the present study differs from those, and how the present methodology brings something different from / completes these studies.

If a clear added value can be defended, then the paper can be published nearly as is. I would add that, if the contribution is an independent evaluation of the likelihood of an ice-free Arctic under 1.5, 2 and 2.5∘C targets, using an alternative method (ensemble vs multi-model vs emulator), I'm quite supportive for the paper to be published, even if the final result duplicates previous findings. Independent, repeated tests are in my view as important as original studies.

In practice, this would probably mean moving material from the end of the paper to the end of the introduction, and complete what is only being suggested at the moment by being more explicit.

Material moved forward to introduction as suggested

More specific comments below.

• The advantages / specificities of the SRM method should be clearer and the reason why it has been chosen as well.

The use of SRM is arguably a plausible mechanism to attain the 1.5C target. It is also a simple mechanism, compared with Sanderson et al (2017), as it requires no new emissions scenario (which would be inconstant with the RCPs). This is an idealised temperature sensitivity study and not suggesting how SRM might be employed. An alternative approach might be to fix CO2 in RCP4.5 when temperatures reach 1.5, 2.0 and 2.5C, but this would leave residual effects from secondary greenhouse gases, aerosols and feedbacks (comparing the 2 methodologies might be an interesting study) SRM is a tried and tested methodology for HadGEM2-ES and the other models of the GeoMIP. We add:

"Here we take a different approach with the intention of assessing if the outcome of three different approaches, including the above, can provide a robust answer to the probability of a seasonally ice-free Arctic at 1.5 and 2.0ºC above pre-industrial. Our methodology is to construct an ensemble of simulations of the CMIP5 model HadGEM2-ES using solar radiation management (SRM) to restrict the global temperature rise. We employ SRM because of its simplicity in requiring a change in just one component to the model, hence maintaining traceability. It is also a plausible scenario, in addition to mitigation, to the 1.5°C target (Sugiyama et al., 2017). This work expands on that of Jones et al (2018) where the SRM methodology is established for HadGEM2-ES. "

• I find the methodology not fully clear. In particular the story of the time dependence of SO2 emissions. Could you illustrate or better describe how SO2 emission depends on time? Is this constant then stabilised? Is it ramped up? Is it non-linear?

The explanation has now been expanded. The injected SO2 volume is time varying to offset the time-varying difference in temperature between RCP4.5 and say the target temperature (say 2.0C). In practice the process (SO2 loading and climate feedbacks) is non-linear with temperature, but for small Delta T this does not matter (as now shown in Figure 3).

• It is well known that the rate of Arctic sea ice decrease depends on mean state, in particular ice volume. Do you expect a model with less volume and the same experimental setup to give higher probability of sea ice volume loss at 1.5◦C ?

A brief comment to this effect, and reference to Bitz (2008), is now included in the conclusions.

• p. 1 l. 23 I would say that there is a net increase in winter growth because ice is thinning (Bitz and Roe, 2004), but I'm not sure which effects dominates. You should come up with more references or more arguments (for instance a mass balance study in CMIP-X).

A comment added to the conclusions refers to the need for such a mass budget analysis.

• p.1 l. 22. "With global, and regional, warming" sounds weird to my ears.

Removed 'and regional'

• p. 1 l. 28 "increased" instead of "increase"

This has been corrected

• p. 1 l. 29. I think the increase in extreme weather due to reduced sea ice is quite challenged, in particular the quite convincing study of Blackport and Kushner J. Clim 2016.

This is still debated e.g Smith et al. (2017) https://doi.org/10.1175/JCLI-D-16-0564.1 and Blackport and Kushner (2017). I think it is still reasonable to say 'may cause'

• p. 2 l. 7. Replace "this is because" by "we make this choice" or reference others to clarify whether you propose this or whether this is standard practice.

This has been corrected as suggested

• p. 2 l. 20. Explain why you use this method.

Have added to the justification at the end of the introduction. Please see response at the start of this response

• One inconsistency is how ◦C is spelt. Sometimes without the ◦, sometimes with space, sometimes not. Make it consistent.

Changed to be consistent throughout

---

## Author Comment (AC2) · 16 Mar 2018

**Response to Referee 2**

We thank the referee for their useful comments

This brief communication shows results on the likelihood of an ice-free Arctic under the low-warming IPCC targets, using experiments with solar radiation management with the HadGEM2-ES model. It shows that limiting the global temperature rise to around 1.5◦C reduces the likelihood of an ice-free Arctic to 0.2%, compared to 43% for warming around 2◦C. This study is interesting and shows similar results to previous work using other models, in particular with the CESM in Sandersen et al. (2017) and the CMIP5 RCPs in Screen and Williamson (2017). This shows that those results are robust across different models. So while the results are not new, they are worthwhile to be published, in particular in light of the upcoming IPCC report on the low warming targets. However, I find the submitted manuscript not publishable in its current form. Even for the category "Brief communication" I find the results section too short – it is shorter than the methods and about the same length as the conclusions and introduction individually. This paper really seems to be pushing the "least publishable unit" into the unacceptable category, and I cannot support that. There is lots of additional analysis that could easily be done with these simulations, even expanding directly on the figures shown. For example Figure 2 is not discussed much and that could be expanded to add new insights not previously shown to this discussion (the spatial aspect of the ice edge under these targets). That said, I have a long list (as long as the paper...) of things that need to be addressed, detailed below. Furthermore, the article has many typos, fuzzy figures, and imprecise and confusing statements and captions. The figures also need to be revised. Overall, I see merit in publishing these results as brief communication, but not in the present form, and therefore recommend major revisions.

- Results section expanded

- Figure quality and information content improved

- More motivation provided

Detailed comments:

Page 1, Line 8-9: Why would we want to "reduce the internal variability" if we want to produce a probability density function of an ice-free state? Is this a typo, and it should be "deduce"?

Clarified with 'improve the signal to noise associated with the internal variability'

Page 1, line 18: Seems to be missing a noun at the end of the sentence? Should be "one such system"

This has been corrected

Page 1, Line 25: Should be "uptake", not "up-take"

This has been corrected

Page 1, Line 28: Should be "increased", not "increase"

This has been corrected

Page 1, Line 3: Why "then" here, doesn't really make sense, as it does not relate to the previous sentence ("Sea ice then hits its smallest extent sometime in September…")

This has been corrected

Page 2, Line 13: two .. at the end of the section

This has been corrected

Page 2, Line 13: I think the introduction needs to reference the of previous work on this topic, and not make it sound like this hasn't been done yet (Screen and Willamson, 2016, Sanderson et al. 2017)

This has been corrected

Page 2, Line 15: Degree symbol is incorrect, should be ∘ not â˚U¸e . This is correct for∘ C, but not for degree related to N/S.

This has been corrected

Page 2, Line 30 to Line 2 Page 3: Is there just one simulation for each RCP? Sounds like it, since just one year is given for reaching these target temperatures.

The appropriate sulphate loading was calculated just once for each target temperature from the mean of the relevant 4-member RCP scenario ensemble. This was initially mentioned P3 line 1+2 but has been moved up and expanded for clarification. Thus, it is indeed the case that ensemble mean is used to determine the dates for the temperature thresholds.

But then it says "For CMIP5 a historical + scenario initial condition ensemble of 4 HadGEM2-ES members was completed.".  So does the ensemble mean of these four cross the threshold at that time? That needs to be made clearer. The figure 1 only shows one line for each RCP. But then says that that is the ensemble mean of 4 RCP members.  But then each member of the 16 member ensemble for reduced temperatures is shown. Why is that done? It makes no sense to me. The 4 ensemble members for each RCP should be shown as well, so it is clear whether the new ensemble is outside the internal variability of the initial RCP. That would help to substantiate the statement on page 3, line 5 that (The resulting ensemble spread in global mean temperature is larger than that for the initial 4-member ensemble).  Not that I doubt that, but it is always nice to see this, and to see how much larger the spread is. Adding the ensemble members for the RCPs would also explain why some of the red lines start well above or below the black line.

As per the reviewer's suggestion, the figures have been updated to show all 4 RCP scenario ensemble members (rather than the mean as before)

Page 2, Line 30 to Line 2 Page 3:  What is the reference period for the 1.5C and 2C and 2.5C warming? That's not mentioned and various papers use different periods.

The reference for pre-industrial is the mean of the parallel pre-industrial control simulation for the period 2005-2100. Pre-industrial refers to 1860 forcing and this has now been added to the ensemble description summaries on page 3.

Page 3, line 3-6: This seems like quite a large perturbation, mixing the ocean initial conditions with an atmosphere that it isn't equilibrated with it. Has this approach been tested before?  I have only heard of others using much smaller initial condition perturbations.

The imposition of a random atmosphere on an ocean state has been demonstrated by Griffies, S. & Bryan, K. Climate Dynamics (1997) 13: 459. https://doi.org/10.1007/s003820050177. The atmosphere equilibrates in days and the ocean loses memory of its initial state within 20 years.

This is now mentioned, and cited, in the text.

Page 3, line 7-10: Why were these target temperatures of 1.3 and 1.7C chosen, when the IPCC target is 1.5C?

The idea was to investigate the region around the 1.5C target to sample for thresholds in the temperature sensitivity. Internal variability was expected to bracket the 1.5 target.

Page 3, Results section:  This is a VERY short result section, shorter than the methods and about the same length as the conclusions.   That seems to me to be very much below the "least publishable unit" standard, and while interesting, I can't support publishing research in such minimal increments.  There is lots of additional analysis that could easily be done with these simulations related to Arctic sea ice to make this acceptable, or even expanding the current description of the results.  I know this is a "Short communication", but even for that I find the results too brief.

The results section has been expanded to include further discussion of the updated Figure 2.

Page 3, line 29: Extra "In" at start of sentence

Corrected

Page 3, line 30: Why is the "quantitative result described here" only listed a paragraph later?  That makes it hard to compare it with the other studies.  Which should be discussed in the introduction already, in my opinion.

Page 4, line 4: Should be "to", not "do"

Corrected

Page 4, Line 1-7:  Didn't Screen and Williamson (2017) look at the probability of just one ice-free Arctic, while here and in Sandersen et al. (2017) the overall probability is assessed?

Screen and Williams use a regression line to produce a single value but then apply Bayesian statistics to obtain a probability.

Page 4, Conclusions in general I think these need to be more carefully written. I don't think the statement "since the Arctic sea ice in CMIP5 models is effectively in equilibrium with the instantaneous global temperature" is defensible, as that would mean that Arctic sea ice in a given

year is dependent on the global temperature.  The cited work shows that this is true in the long-term sense, but not for year-to-year variability. Furthermore, this doesn't apply to all properties of Arctic sea ice, so this needs to be more precisely formulated (September, extent).

Agreed. This comment and associated references has been removed

Generally: Please use a consistent number of significant digits, and not 1.5C and 2.5C but then 2C.

Consistency corrected throughout

References: Please check these for inconstant formatting – not my job as a referee to fix those

Done

Figure 1:  As mentioned earlier, this figure should either show the ensemble mean or the individual members, not a mix of both.  Unless there is a good reason for that, but none is articulated. Furthermore, the figure is fuzzy, so it doesn't seem to be saved in a vector format. Need to be switched out to be acceptable.

Figure 2: It is not clear to me what this figure really shows. First of all, this figure does not show the "spatial pattern of the Arctic sea ice extent", but the "spatial pattern of the sea ice edge". Which is only the 15% contour, and hence is very noisy. Furthermore, it is unclear to me what both the red and black lines are. It says early on that it shows the "mean of years 2080-2099", for certain temperature thresholds in certain simulations. But then it goes on to say it shows "the mean (years 2006-2025) of the four" RCPs.  I also don't understand "starting at +1.5∘C above preindustrial in RCP2.6 (left), +2∘C in RCP4.5 (centre) and +2.5∘C in RCP4.5 (right)."  This clearly needs to be rewritten to make sense to a reader who hasn't made the figure.  It says the red lines are for the 16 individual ensemble members, but those are not the RCPs. Going back to how the ensembles are defined, I think that's what the red lines show, but then these are not for temperatures 1.5C, 2.0C and 2.5C, but for 1.5C, 1.3C and 1.7C? Panels also should have labels (a) and (b) and (c) for ease of reading/referencing. Furthermore, based on what I see, the black line in each panel is different.  I assume that is because it is for the different temperature thresholds in the two RCPs mentioned above, and probably for 2006-2025, as they look like present-day. But why use different baselines for each, and what is the third one, since there are only two RCPs? That all makes no sense to me. It would make more sense to use the present-day period for all of them, potentially both for the model and observations, to show how realistic the model is. And then show the ice edge for the 2080-2099 for each of the three ensembles.  But since their temperatures aren't that different, and are lumped together in the next figure, why they should be shown is unclear.  To show the difference between the ones that end at 1.5 and 1.3 degree C versus ensemble 3 that ends at 1.7 degree C?

Figure 3:  This figure is also fuzzy and out of focus.  It needs to include a higher resolution figure to be considered for publication.  Furthermore, it is unclear what exactly it shows.  The caption needs to state what the size of the boxes is (+/- 0.1 degree C around the mentioned thresholds, according to my reading of the graph), as it does not show the probability right at the quoted temperature thresholds.  Otherwise, this figure is the most interesting one. I do wonder how it compares to the transient RCPs with that model though, and hence how much it adds to just using those.  Can those be added here?  They all cross the same temperature range, so could be included in the PDF. Currently it seems to only include the 3 ensembles with 16 members each (3*16=48).  That leads to

many fewer members that are in the 2 degree warming box than in the 1.5 degree warming box, and hence could be influencing the probability distributions shown in panel 2. Adding the RCPs in here would help to rectify this. Panels also should have labels (a) and b.

The figures have all been updated in line with all the above suggestions

---

## Referee Comment (RC3) · Anonymous Referee #3 · 28 Mar 2018

The submitted paper discusses the response of Arctic Sea ice September coverage at global mean temperatures of 1.5 or 2 degrees Celcius above pre-industrial, the targets referred to in the Paris climate agreement.

In contrast to previous studies, which have focussed on scenarios which achieve these targets through greenhouse gas mitigation alone, the present study considers joint mitigation and solar radiation management (SRM) to achieve global mean temperature goals.

The study is potentially interesting, but does not address the most interesting issue which could potentially be discerned from this dataset: is there a difference in the

projected avoided sea-ice loss which can be obtained through solar radiation management, compared to greenhouse gas mitigation alone?

Firstly - the authors have considered only one target in their geo-engineering experiment: the global mean temperature, and the authors have used globally uniform sulphate distributions to represent their SRM. It has long been noted that such compensation of uniform sulphate increase, whose effect peaks in the tropics combined with increased $CO_2$, whose effect peaks at the pole - results in significant warming at the poles relative to the $CO_2$ mitigation case (Ricke 2010). This would imply that the author's estimates of ice distribution at 1.5 or 2 degrees are likely to show more loss than a pure mitigation case. This is undiscussed in the paper - and is a central point.

Moreover, recent studies have highlighted that targeted injection patterns can mitigate the polar warming effect (Kravitz 2017, Modak 2013) by increasing choosing injection sites which increase the relative sulphate loading over the poles or summer hemisphere. Even if the authors' model is not capable of resolving interactive aerosols, a non-uniform sulphate loading distribution could quantify the efficacy of such approaches for sea ice conservation.

A clear possibility here is to quantify minimum sea ice cover not just as a function of global mean temperature - but as a function of forcing type and transient forcing history (is there any detectable lag in the response of sea ice to falling temperatures as the sulphate loading is increased?).

This is an interesting dataset, but it has been interpreted as a straightforward assessment of climate at 1.5 and 2 degrees, although there are strong reasons to believe that the geoengineered climates considered here would be unlike those observed at global mean temperatures of 1.5 or 2 degrees during a conventional RCP. The paper should acknowledge this, and consider more deeply how climate targets achieved using SRM differ from those achieved using mitigation.

Minor Issues:

The injection quantities use information derived from the multi-model mean - which is a piece of information which would not be known in the real world. This should be acknowldged

There are multiple typos. Please proof read before resubmission.

Ricke, K. L., Morgan, M. G., & Allen, M. R. (2010). Regional climate response to solar-radiation management. Nature Geoscience, 3(8), 537.

Kravitz, Ben, Douglas G. MacMartin, Michael J. Mills, Jadwiga H. Richter, Simone Tilmes, Jean‐Francois Lamarque, Joseph J. Tribbia, and Francis Vitt. "First simulations of designing stratospheric sulfate aerosol geoengineering to meet multiple simultaneous climate objectives." Journal of Geophysical Research: Atmospheres 122, no. 23 (2017).

Modak, A., and G. Bala. "Sensitivity of simulated climate to latitudinal distribution of solar insolation reduction in SRM geoengineering methods." Atmos Chem Phys Discuss 13 (2013): 25387-25415.

---

## Author Comment (AC3) · 4 Apr 2018

We thank the referee for their insight into the Geoengineering aspect of the paper

The submitted paper discusses the response of Arctic Sea ice September coverage at global mean temperatures of 1.5 or 2 degrees Celsius above pre-industrial, the targets referred to in the Paris climate agreement.

In contrast to previous studies, which have focussed on scenarios which achieve these targets through greenhouse gas mitigation alone, the present study considers joint mitigation and solar radiation management (SRM) to achieve global mean temperature goals.

The study is potentially interesting, but does not address the most interesting issue which could potentially be discerned from this dataset: is there a difference in the projected avoided sea-ice loss which can be obtained through solar radiation management, compared to greenhouse gas mitigation alone?

Firstly - the authors have considered only one target in their geo-engineering experiment: the global mean temperature, and the authors have used globally uniform sulphate distributions to represent their SRM. It has long been noted that such compensation of uniform sulphate increase, whose effect peaks in the tropics combined with increased $CO_2$, whose effect peaks at the pole - results in significant warming at the poles relative to the $CO_2$ mitigation case (Ricke 2010). This would imply that the author's estimates of ice distribution at 1.5 or 2 degrees are likely to show more loss than a pure mitigation case. This is undiscussed in the paper - and is a central point.

Moreover, recent studies have highlighted that targeted injection patterns can mitigate the polar warming effect (Kravitz 2017, Modak 2013) by increasing choosing injection sites which increase the relative sulphate loading over the poles or summer hemisphere. Even if the authors' model is not capable of resolving interactive aerosols, a non-uniform sulphate loading distribution could quantify the efficacy of such approaches for sea ice conservation.

A clear possibility here is to quantify minimum sea ice cover not just as a function of global mean temperature - but as a function of forcing type and transient forcing history (is there any detectable lag in the response of sea ice to falling temperatures as the sulphate loading is increased?).

This is an interesting dataset, but it has been interpreted as a straightforward assessment of climate at 1.5 and 2 degrees, although there are strong reasons to believe that the geoengineered climates considered here would be unlike those observed at global mean temperatures of 1.5 or 2 degrees during a conventional RCP. The paper should acknowledge this, and consider more deeply how climate targets achieved using SRM differ from those achieved using mitigation.

We would agree that there is more that could be looked at in these experiments related to geoengineering, much of which could be made relevant to the impacts on the cryosphere. Another paper, Wiltshire et al. (in preparation), looks at the impacts of different pathways to a target global temperature through geoengineering and mitigation. In the longer term the impacts of different injection patterns may also be investigated, probably using our CMIP6 model which has a well resolved stratosphere and complex chemistry scheme. However, our choice in the format of a Brief Communication, is to focus on a single topic; the response of the Arctic sea ice to 1.5C and 2C global

temperatures. The SRM approach is different than other published methodologies and in that context the SRM is the means to an end in our case using existing methodologies and simulations.

Figure 3 now shows the RCP4.5 (mitigated to 2.8 degrees C) and RCP2.6 (mitigated to 2 degrees C) scenarios as well as the SRM simulations. Reference is made to Kravitz et al (2017) and Jones et al. (2018), which remarks on the reduction of polar amplification using SRM. We note that at 1.5 C there is a difference in ice cover in the transient simulations over those of the SRM ensembles, but it is not statistically significant (also mentioned in Results section). In essence the Arctic sea ice appears to simply be responding to global temperature in a time averaged sense.

Minor Issues:

The injection quantities use information derived from the multi-model mean - which is a piece of information which would not be known in the real world. This should be acknowledged.

Now mentioned in the Methods section.

There are multiple typos. Please proof read before resubmission.

Many typos corrected.

Ricke, K. L., Morgan, M. G., & Allen, M. R. (2010). Regional climate response to solar-radiation management. Nature Geoscience, 3(8), 537.

Kravitz, Ben, Douglas G. MacMartin, Michael J. Mills, Jadwiga H. Richter, Simone Tilmes, Jeanâ A R Francois Lamarque, Joseph J. Tribbia, and Francis Vitt. "First simulations of designing stratospheric sulfate aerosol geoengineering to meet multiple simultaneous climate objectives." Journal of Geophysical Research: Atmospheres 122, no. 23 (2017).

Modak, A., and G. Bala. "Sensitivity of simulated climate to latitudinal distribution of solar insolation reduction in SRM geoengineering methods." Atmos Chem Phys Discuss, 13 (2013): 25387-25415.

---

## Author Response (AR2)

**Response to REFEREE-2 (marked-up version of paper at bottom)**

We thank the referee for the comments, and in light of the new publications on this subject have truncated discussion of previous methodologies to limit text and expound the unique SRM focus of this paper.

This is my second review of the brief communication, "The significance for the IPCC targets of 1.5°C and 2.0°C temperature rise for an ice-free Arctic." This version of the manuscript is much improved, but unfortunately, I still don't think it is ready for publication, as some of the newly added text is confusing, some of it is incorrect, and the figures are still all out of focus and not acceptable (despite a reply to review that stated these figures were updated in response to review). Figures should be vector graphics (eps or pdf) to avoid them being this pixelated at a normal print and screen resolution (100%). Furthermore, since the first review, three new papers on this exact same subject have now appeared (see below for the new references), with another in open review in earth system discussions (Iversen et al., see below). Those need to discussed in a further review cycle (at least the published ones, the under review one maybe not, not sure on the policy for those articles in open review). I don't think these new studies preclude the publication of this paper, as they use different models and methodologies, and the solar radiation management approach is unique to this study. And it is very interesting that the results agree quite well. But these studies need to be discussed, and the introduction needs to be rephrased in order to reflect that there are previous answers to this question, and that the main contribution here is to look at the impact of solar radiation management, as alternative to reduced emissions, as mentioned in the reply to review. I think this will actually help the article to be more interesting and significant.

We now refer to the latest 3 publications, and as a result do not specifically mention differences in methodology. Instead, they are grouped together as mitigation studies for reference to the specifically geo-engineering study of this paper. The refocus of the paper specifically on SRM has necessitated a major reworking of the manuscript, although the figures and basic results remain the same.

Specific comments:

Page 1,Line 1: I would recommend reconsidering the title to better reflect the unique aspects of this study. Maybe "Brief Communication: Probabilities of an ice-free Arctic for limiting warming to 1.5C and 2.0C though solar radiation management"

The title is changed to "Solar Radiation Management not as effective as mitigation for Arctic sea-ice loss in hitting the 1.5°C and 2°C COP climate targets."

This is to emphasise a caution that SRM is not a cure-all

Page 1, Line 8-9: It is unclear to me what "is used to improve the signal to noise associated with the internal variability" means, so that is not good in an Abstract, which should convey the main message clearly and concisely.

The abstract is rewritten to reflect the changed storyline of the paper. This phase above is removed.

Page 1, Line 9-10: I don't think this statement is correct, based on the analysis presented: "It is found that the continuing loss of Arctic sea ice can be halted if the Paris Agreement temperature goal of 1.5oC is achieved". The paper only assessed September ice extent, whereas this statement implies any ice loss, which can include winter month and/or ice volume. Also, arguably also the 2C stabilization stops the ice-loss (in September), but at a lower level. So this statement is just not correct and can't be in the Abstract. Instead there should be a statement on the actual finding, as stated in the conclusions (incorrectly, as increased rather than decreased, see comment later) and supported by the analysis presented: "The risk for a seasonally ice-free Arctic is reduced for a target temperature for global warming of 1.5°C (0.1%) compared to 2.0°C (42%)."

The suggested phrase has been adopted

Page 1, line 18-19: Given that there are now three more 2018 published studies on this subject, and a News and Views text by Screen (2018), in addition to the cited two 2017 studies (Screen and Williamson and Sanderson et al), "Here we investigate if Arctic sea ice cover is one such system." Really doesn't seem appropriate anymore. All of these studies have shown that the Arctic sea ice cover is such a system. So while this may have been the initial motivation, I would suggest re-casting it in terms of whether global temperature control through solar radiation management also shows such large impacts between 1.5C and 2.0C for Arctic sea ice, or whether solar radiation management would lead to different results. That requires re-writing large parts of the initial paragraph, but I think it will make the paper much stronger.

The paper has been heavily re-written throughout to focus on the geoengineering aspect

Page 2, line 11-12: This is incorrect "Sanderson et al., 2017) used a climate model emulator, calibrated against CESM1, to produce simulations at constant 1.5°C and 2.0°C". They used a climate model emulator to design emission scenarios that lead to CESM1 simulations at constant 1.5°C and 2.0°C. But the results shown are from the CESM1, not from an emulator. This needs to be correctly stated.

Statement is removed as commentary on the individual techniques is superseded by the range of new papers on the topic. Instead we reference the Screen (2018) commentary.

Page 2, line 12-14: This is also incorrect "Another study (Screen & Williamson, 2017) used all the CMIP5 model simulations and regressed the September sea ice extent for 1.5°C and 2.0°C against the 2007-2016 mean sea ice extent. Bayesian statistics were then used to estimate the probability of an ice-free Arctic for 1.5°C or 2.0°C.". Screen & Williamson used CMIP5 simulations until the year before they first reached a global warming of 1.5°C and 2.0°C and regressed those against the 2007-2016 mean sea ice extent.

With the additional 3 studies it no longer makes sense, to review the individual methodologies of the 5 existing papers, so this section is removed.

Page 2, line 14: A discussion of the new three studies need to go here.

The reorganisation of the paper to address geoengineering has required all these studies grouped together as mitigation simulations.

Page 2, line 15: Now of six different approaches

New studies referenced

Page 3, line 6-7: I know this was added in response to reviewer 3 (The use of an ensemble mean will not be available in reality to plan implementation of SRM.), but without context, this is not very useful and should be expanded upon, so the reader can understand why this is here, without having to read the reviews.

The statement about '..in reality to plan implementation of SRM' is incongruous with the rest of the paper, and the line has been removed This paper is not about discussing the practical implementation of SRM – there have been other papers on such.

Page 3, line 12-16: This is an important and very interesting point, but I needed to read it three times to get it. Please re-write it in shorter, more precise sentences, so it is easier to follow.

Rewritten slightly.

Page 4, line 17-19: In order to really understand Figure 3b, one need to know how many years contribute to each of the pdfs. I would recommend adding the temperature bands in Figure 1, so it is clear how many years of ensemble 1, 2, and 3 end up in those bands and are used for the assessment of these probabilities. Because Sigmond et al showed that this probability increases the longer the sample period is, as also discussed by Screen (2018).

As Jahn suggests this is related to the ensemble size and hence the sampling due to the members rather than the length of time. If it were just due to length of time then the probability would continue to increase beyond the first 10 years in Sigmond et al. With an ensemble of 48 members in these experiments it is clear from Fig 3 that the spread is far greater than for the four member RCP ensemble. We now mention in the text the sample size of the histograms in Fig 3 (1000 at 1.5C and 300 at 2.0C). An increase of variability in time could also occur if the ice continues to thin, perhaps in response to an initial shock at the start of the simulations (unless the simulations have started from a spun-up state).

In addition, (or alternately, if those boxes make Figure 1 too busy), the authors should mention how many years are in each of the bands, and from which ensembles they are (RCP2.6, RCP4.5, ensemble-1, ensemble-2, ensemble-3). This is important to understand the methodology.

We cannot distinguish, statistically (as a function of global mean temperature – noting that global temperature drifts in ensembles 2 & 3)), any difference in the standard deviation between the three ensemble categories, so we have lumped all the ensemble (and RCP) members together. To avoid crowding figure 3 we have mentioned the number of years (sample size) in the results section of the text.

Page 4: Line 21: This is the central finding, but there is a typo: The risk is significantly DECREASED, not increased as stated, for a target temperature of 1.5C.

Thanks for spotting this. Text now says "A significantly reduced risk…"

Page 4, Line 22: It is unclear to me how this statement fits here, as the first sentence isn't about timing, but general probabilities. So "Another climate model, with a thicker Arctic sea ice in its mean state, would be expected to produce a later date for an ice-free Arctic (Bitz, 2008) " doesn't fit here. Please either remove it or add a sentence that actually refers to the date of ice-free conditions. But that wasn't analyzed here, so really I think this needs to go.

I agree, models can vary in the ice-free date but that could be due to a variety of reasons such as climate sensitivity, polar amplification and initial state. Although polar amplification is now mentioned, in relation to SRM, we do not infer anything that might relate to different models. The sentence is removed.

Page 4, line 23-24: Screen and Williamson assessed the probability of any ice-free conditions before reaching 1.5C and 2.0C, not the probability in a given year, as is done here, and was done in Sanderson et al. (2017). That needs to be clarified. See Screen 2018 (see below for ref) for an explanation of the differences of those probabilities.

Methodologies of different papers no longer discussed here. Section removed.

Page 4, Line 25: Sanderson et al did not use a climate model emulator for these results, as stated here incorrectly, but used a climate model emulator to develop the emission scenarios that would lead to stabilized warming at 1.5C and 2.0C in the CESM1. That's clearly explained in Sanderson et al.

Methodologies of different papers no longer discussed here. Section removed.

Figures: All figures are still low quality and out of focus. They need to be vector graphics.

I have no problem with the figures (resolution) when I download the paper from the Cryosphere. In any case the final graphics will be high resolution.

Minor comments:

Figure 2: As noted last time, figure 2 does not show the "spatial pattern of the Arctic sea ice extent", but the "spatial pattern of the sea ice edge". Other changes were made, but this one wasn't.

Changed as suggested

Figure 3: The caption now better explains the figure. But the newly added line for ice-free is a purple line, not a dark blue line.

Caption changed to 'purple'

Page 3, line 18: As for the next two lines, this should read: Ensemble-1 : starts at 1.5C on RCP2.6 and levels out at 1.5C above pre-industrial

Implemented changes as suggested

Page 3, line 29: "with ice retreating further in the Atlantic sector" In which ensemble, ensemble-1 or ensemble-2?

This refers to HadGEM2-ES in general (now clarified in the text) and is associated with the modelled ice rheology as well as weak (compared with observations) atmospheric blocking in the North Pacific.

Page 1, line 21: This is not a "salinity freezing point", but "salinity-dependent freeze point" or "freezing point of the ocean".

Changed to 'freezing point of the ocean'

Page 1, line 23: Missing "a" before "warmer atmosphere" The revised version of "The significance for the IPCC targets of 1.5°C and 2.0°C temperature rise for an ice-free Arctic." addresses some, but not all of the concerns I had with the original paper.

Inserted 'a'

**Response to REFEREE 3 (Marked up version of paper at bottom)**

We thank the referee for the insightful comments to improve this brief communication. We have endeavoured to incorporate all suggestions and yet limit the text to maintain the Brief Communication

My major concern was that an SRM experiment was being used as a proxy for climate impacts at 1.5 and 2 degrees, with particular reference to those temperatures as described in the Paris Agreement. This remains the case with this version, and it is troubling for 2 reasons:

The high latitude warming at a given stabilization level will be greater in a simulation where some greenhouse gas forcing is offset with SRM. This is supported by the paper's new Figure 3a, which clearly shows that the mean sea ice extent at 1.5 degrees in the geoengineered simulations is less than that seen in the mitigation-only experiments (i.e. the mean of the distribution of black points is less than the mean of the distribution of red points at the 1.5 degree warming level). However, the authors argue the opposite - that their analysis demonstrates that the SRM simulation is a good proxy for mitigation only experiments.

The paper is completely re-written with a focus on SRM. New analysis has shown that, as expected, the polar amplification under SRM is higher and thus the rate of sea ice loss is increased. We do not have a sufficiently large ensemble of RCP scenarios to determine what the probabilities for ice-free at 1.5 and 2.0C would be if no SRM were used (in the same model).

This is potentially misleading to the community. The Paris Agreement targets are usually interpreted as a mitigation goal - not as targets for SRM geo-engineering. This paper is too quick to dismiss the

difference between the climates which would be achieved in the two cases, despite the fact that their results show clear differences.

See above

As I said in my initial report, this paper would be quite acceptable if it acknowledged that there are differences in SRM simulations and mitigation only experiments (and categorized them in detail for Arctic sea-ice) - but the paper still reads as if SRM and mitigation are interchangeable. This is a strong statement, with huge policy implications - and it is not even supported by the author's own results.

This comment, and the focus of the paper on SRM, has led to a more though statistical analysis of the impact of mitigation vs SRM. Despite the narrow temperature range (+- 0.1C) around the target temperatures of 1.5 and 2.0C, the sample size is large and the difference is statistically significant. The results show a higher polar amplification for SRM than Mitigation, for the same global mean temperature, consequently the sea ice area is less. Clearly, although SRM can make the world cooler than 1.5C (very difficult for mitigation) this is not the purpose of this brief communication.

Minor Issues:

- Remove apostrophes in "RCP's"

Done

- Section 3, last but 1 para " SRM itself is not having an effect on the Arctic sea ice " - this statement is not supported by the plot, the two distributions are clearly distinguishable at the 1.5 degree threshold.

- "Sandersen et al. (2017)" -> "Sanderson et al. (2017)" throughout

Done

- Section 4 - to say Sanderson et al used a climate model emulator is misleading - the emulator was only used to produce the emission pathways. The relationship between global mean temperature and arctic sea ice found in that paper was derived from a full GCM.

Description removed as it is no longer necessary to highlight differences in the methodologies

- Section 4 - 2nd para - "independent" is a strong word with statistical implications which do not apply here. Consider using simply "different"

Changed to 'different'

**Response to Editor Comment**

**In addition, the issue that most of the current GCM-based simulations underestimates the currently observed sea ice decline in Arctic is still not yet discussed as well as a validation of the model used over current climate/sea ice trend.**

It is generally agreed that about half the current observed sea ice trend is associated with multi-decadal variability. As such a model only needs to demonstrate that it can reproduce 30/20/10 year trends, compatible with observations within an ensemble of historical simulations. There are plenty of papers that demonstrate this principle. Thus, I don't feel this a fruitful avenue of discussion in the paper.

https://www.nature.com/articles/nclimate2483

https://www.sciencedirect.com/science/article/pii/S092181811530093X?via%3Dihub

https://agupubs.onlinelibrary.wiley.com/doi/abs/10.1002/2016GL070067
* * *
**Marked up version of manuscript with major changes in red**

Brief Communication: Solar Radiation Management not as effective as $CO_2$ mitigation for Arctic sea-ice loss in hitting the 1.5°C and 2°C COP climate targets.

Jeff K. Ridley, Edward W. Blockley

Met Office, Exeter, EX1 3PB, UK

*Correspondence to*: Jeff Ridley (jeff.ridley@metoffice.gov.uk)

**Abstract.** An assessment of the risks of a seasonally ice-free arctic at 1.5 and 2.0°C global warming above pre-industrial is undertaken using model simulations with solar radiation management to achieve the desired temperatures. An ensemble, of the CMIP5 model HadGEM2-ES uses solar radiation management (SRM) to achieve the desired global mean temperatures. It is found that the risk for a seasonally ice-free Arctic is reduced for a target temperature for global warming of 1.5°C (0.1%) compared to 2.0°C (42%), in general agreement with other methodologies. The SRM produced more ice loss, for a specified global temperature, than for $CO_2$ mitigation scenarios, as SRM produces the higher polar amplification. .

**1 Introduction**

The 21st Conference of Parties (COP) to the UN Framework Convention on Climate Change held in Paris in 2016 made a commitment to limiting global-mean warming since the pre-industrial era to well below 2.0°C and to pursue efforts to limit the warming to 1.5°C (UNFCCC, 2015). The 1.5 °C target reflects a threshold at which the likely local impacts of climate change are beyond the ability of society to cope with. This is especially applicable to the small island states that are susceptible to sea-level rise, ground-water salinification and loss of coral reefs. One such risk is the loss of Arctic sea ice, for which previous studies (Sanderson et al., 2017; Screen & Williamson, 2017; Jahn, 2018; Niederdrenk & Notz, 2018; Sigmond et al., 2018) used a number of methodologies with various climate models under $CO_2$ mitigation scenarios. The findings are broadly similar, that there is a low

chance of an ice–free Arctic if global temperatures are limited to 1.5°C and a moderate chance at 2°C.

It has been suggested that geoengineering, otherwise known as solar radiation management (SRM), may be a stopgap measure to halt these impacts,  stabilising Earth's temperature at 1.5 K, before $CO_2$ mitigation can take effect (Chen & Xin, 2017). Here we evaluate the impact of SRM on Arctic sea ice decline and compare with mitigation methods alone, through the implementation of SRM, in our climate model HadGEM2-ES. We use the SRM strategy of stratospheric aerosol injection, which mimics large volcanic eruptions (Crutzen, 2006).

Arctic sea ice area declines and thins in summer due to surface melting and solar absorption in open water resulting in warming and melting at the ice base. In the absence of incoming solar radiation, ice thickens and spreads in winter caused by heat loss to the colder atmosphere cooling the ocean to its freezing point, 
[revised manuscript text omitted]

---

## Author Response (AR3)

The third version of this manuscript is finally in good shape (beside the figures). By refocusing the manuscript on the SRM difference to CO2 mitigation, the paper makes a clear and novel contribution to the discussion, while citing most of the recent relevant literature (should add Tilmes et al 2014 (GRL), which assessed the effect of global versus regional shortwave radiation management on Arctic sea ice.) I can now almost recommend publication, provided final edits are done as outlined below, including finally replacing the figures with higher quality figures, as requested twice already to no anvil.

As the wrong manuscript file was submitted, I can only comment on the version included in the reply to review, assuming this is the version the authors wanted to submit. Unfortunately, that has no line numbers, so I can't refer to the exact place in the paper in my comments below.

End of Abstract: Should be "produces a higher amplification".

Corrected

Introduction 3rd paragraph: I think this whole paragraph should go, as it isn't directly related to anything that is later discussed, and it makes the Introduction longer than the results section, which is odd for a paper.

Removed paragraph as suggested reducing word count.

Introduction 3rd paragraph (if kept): Missing comma and please check the grammar in this sentence after "The thinner the ice the less that survives the summer melt and consequently the area of perennial ice declines"

Introduction 3rd paragraph the Introduction is longer than the results section.: Should be "ice-albedo feedback"

3rd paragraph removed

Introduction 4th paragraph: Please take out "The current minimum…", as this statement is unnecessary (never referred to again), is based on daily sea ice extent (when the rest is based on monthly means), and there is no pre-industrial or 1980s reference value given, so just one number does not support the following statement of "Such a sharp drop off in sea ice..". This fits much better if it follows the 11% decline per decade in the sentence before the one I would recommend taking out.

Agreed, the sentence has been removed.

Introduction 5th paragraph: The coastal erosion is not just a problem if the ice loss contributes to increase extreme weather in mid-latitudes, but in general as there is a longer open-water season in the Arctic. So this should be "Furthermore,", not "Thus", which implies a causal link to the mid-latitude weather changes.

Agreed. It was not the attention to infer causality. "Thus " replaced with "Furthermore"

Introduction 6th paragraph: Should be "It has been suggested", not "It is suggested" at start of red text. In this discussion, or later, the Tilmes et al 2014 (GRL) paper should also be cited, which assessed the effect of global versus regional shortwave radiation management on Arctic sea ice.

Corrected as suggested. The reference to Tilmes et al has been included in the conclusions when asymmetric SRM is introduced.

Conclusions: two .. after "SMR and RCP..

Corrected

Conclusions: Last sentence should be clarified, and include the cause of this: "Here we show that SRM is not as effective as conventional mitigation in reducing Arctic sea-ice loss, due to a higher polar amplification for SRM for the same amount of global warming." As this is the last sentence, having it be easy to follow and strong is important. The suggested rewrite also sounds less like an endorsement of SRM, which the authors made clear in another place is not what they are doing.

Thanks. Your better final sentence has been adopted

References: Several references appear out of order (Overland in between McLaren and Ming; Rayner between Sanderson and Shepherd), one is cut off (ure, N, Saches, T., Helm, V., and Fritz, M.). So extra care should be taken to ensure all references cited are included, in the correct order, and without mistakes.

References checked and corrected.

Figures: These figures are still of very bad quality and can not be published like this. The authors replied to my previous comment on that saying "I have no problem with the figures when I downloaded the paper from the Cryosphere. In any case the final graphics will be high resolution". I have a problem with the figures, they are pixelated in the versions I have, making the text in them hard to read (like legend text) and the lines blur together. If high quality versions exist, why not include them now, after two requests for that?

EPS versions of the figures exist but cannot be included in word. They can be separately downloaded to Cryosphere as required, but no such capability currently exists